# Breaking the Curse of Horizon: Infinite-Horizon Off-Policy Estimation

Qiang Liu
The University of Texas at Austin
Austin, TX, 78712
lqiang@cs.utexas.edu

Lihong Li
Google Brain
Kirkland, WA, 98033
lihong@google.com

Ziyang Tang
The University of Texas at Austin
Austin, TX, 78712
ztang@cs.utexas.edu

Dengyong Zhou
Google Brain
Kirkland, WA, 98033
dennyzhou@google.com

## Abstract

We consider off-policy estimation of the expected reward of a target policy using samples collected by a different behavior policy. Importance sampling (IS) has been a key technique for deriving (nearly) unbiased estimators, but is known to suffer from an excessively high variance in long-horizon problems. In the extreme case of *infinite*-horizon problems, the variance of an IS-based estimator may even be unbounded. In this paper, we propose a new off-policy estimator that applies IS *directly* on the stationary state-visitation distributions to avoid the exploding variance faced by existing methods. Our key contribution is a novel approach to estimating the density ratio of two *stationary state distributions*, with *trajectories* sampled from only the behavior distribution. We develop a mini-max loss function for the estimation problem, and derive a closed-form solution for the case of RKHS. We support our method with both theoretical and empirical analyses.

## 1 Introduction

Reinforcement learning (RL) [36] is one of the most successful approaches to artificial intelligence, and has found successful applications in robotics, games, dialogue systems, and recommendation systems, among others. One of the key problems in RL is policy evaluation: given a fixed policy, estimate the average reward garnered by an agent that runs this policy in the environment. In this paper, we consider the off-policy estimation problem, in which we want to estimate the expected reward of a given target policy with samples collected by a different behavior policy. This problem is of great practical importance in many application domains where deploying a new policy can be costly or risky, such as medical treatments [26], econometrics [13], recommender systems [19], education [23], Web search [18], advertising and marketing [4, 5, 38, 40]. It can also be used as a key component for developing efficient off-policy policy optimization algorithms [7, 14, 18, 39].

Most state-of-the-art off-policy estimation methods are based on importance sampling (IS) [e.g., 22]. A major limitation, however, is that this approach can become inaccurate due to the high variance introduced by the importance weights, especially when the trajectory is long. Indeed, most existing IS-based estimators compute the weight as the product of the importance ratios of many steps in the trajectory. Variances in individual steps accumulate *multiplicatively*, so that the overall IS weight of a random trajectory can have an exponentially high variance to result in an unreliable estimator. In the extreme case when the trajectory length is infinite, as in infinite-horizon average-reward problems, some of these estimators are not even well-defined. Ad hoc approaches can be used, such as truncating

the trajectories, but often lead to a hard-to-control bias in the final estimation. Analogous to the well-known "curse of dimensionality" in dynamic programming [2], we call this problem the "curse of horizon" in off-policy learning.

In this work, we develop a new approach that tackles the curse of horizon. The key idea is to apply importance sampling on the *average visitation distribution* of single steps of state-action pairs, instead of the much higher dimensional distribution of whole trajectories. This avoids the cumulative product across time in the density ratio, substantially decreasing its variance and eliminating the estimator's dependence on the horizon.

Our key challenge, of course, is to estimate the importance ratios of average visitation distributions. In practice, we often have access to both the target and behavior policies to compute their importance ratio of an action conditioned on a given state. But we typically have *no* access to transition probabilities of the environment, so estimating importance ratios of state visitation distributions has been very difficult, especially when only off-policy samples are available. In this paper, we develop a mini-max loss function for estimating the true stationary density ratio, which yields a closed-form representation similar to maximum mean discrepancy [9] when combined with a reproducing kernel Hilbert space (RKHS). We study the theoretical properties of our loss function, and demonstrate its empirical effectiveness on long-horizon problems.

## 2   Background

**Problem Definition**   Consider a Markov decision process (MDP) [31] $M = \langle \mathcal{S}, \mathcal{A}, r, \boldsymbol{T} \rangle$ with state space $\mathcal{S}$, action space $\mathcal{A}$, reward function $r$, and transition probability function $\boldsymbol{T}$. Assume the environment is initialized at state $s_0 \in \mathcal{S}$, drawn from an unknown distribution $d_0(\cdot)$. At each time step $t$, an agent observes the current state $s_t$, takes an action $a_t$ according to a possibly stochastic policy $\pi(\cdot|s_t)$, receives a reward $r_t$ whose expectation is $r(s_t, a_t)$, and transitions to a next state $s_{t+1}$ according to transition probabilities $\boldsymbol{T}(\cdot|s_t, a_t)$. To simplify exposition and avoid unnecessary technicalities, we assume $\mathcal{S}$ and $\mathcal{A}$ are finite unless otherwise specified, although our method extends to continuous spaces straightforwardly, as demonstrated in experiments.

We consider the *infinite horizon* problem in which the MDP continues without termination. Let $p_\pi(\cdot)$ be the distribution of trajectory $\boldsymbol{\tau} = \{s_t, a_t, r_t\}_{t=0}^\infty$ under policy $\pi$. The expected reward of $\pi$ is

$$R_\pi := \lim_{T \to \infty} \mathbb{E}_{\boldsymbol{\tau} \sim p_\pi}[R^T(\boldsymbol{\tau})], \qquad R^T(\boldsymbol{\tau}) := (\sum_{t=0}^T \gamma^t r_t)/(\sum_{t=0}^T \gamma^t),$$

where $R_\pi^T(\boldsymbol{\tau})$ is the reward of trajectory $\boldsymbol{\tau}$ up to time $T$. Here, $\gamma \in (0, 1]$ is a discount factor. We distinguish two reward criteria, the average reward ($\gamma = 1$) and discounted reward ($0 < \gamma < 1$):

$$\textit{Average:} \quad R(\boldsymbol{\tau}) := \lim_{T \to \infty} \frac{1}{T+1} \sum_{t=0}^T r_t, \qquad \textit{Discounted:} \quad R(\boldsymbol{\tau}) := (1 - \gamma) \sum_{t=0}^\infty \gamma^t r_t.$$

where $(1 - \gamma) = 1/\sum_{t=0}^\infty \gamma^t$ is a normalization factor. The problem of *off-policy value estimation* is to estimate the expected reward $R_\pi$ of a given *target* policy $\pi$, when we only observe a set of trajectories $\boldsymbol{\tau}^i = \{s_t^i, a_t^i, r_t^i\}_{t=0}^T$ generated by following a different *behavior* policy $\pi_0$.

**Bellman Equation**   We briefly review the Bellman equation and the notation of value functions, for both average and discounted reward criteria. In the discounted case ($0 < \gamma < 1$), the value $V^\pi(s)$ is the expected total discounted reward when the initial state $s_0$ is fixed to be $s$: $V^\pi(s) = \mathbb{E}_{\boldsymbol{\tau} \sim p_\pi}[\sum_{t=0}^\infty \gamma^t r_t \mid s_0 = s]$. Note that we *do not* normalize $V^\pi$ by $(1 - \gamma)$ in our notation. For the average reward ($\gamma = 1$) case, the expected average reward does not depend on the initial state if the Markov process is ergodic [31]. Instead, the value function $V^\pi(s)$ in the average case measures the *average adjusted* sum of reward: $V^\pi(s) = \lim_{T \to \infty} \mathbb{E}_{\boldsymbol{\tau} \sim p_\pi}[\sum_{t=0}^T (r_t - R_\pi)|s_0 = s]$. It represents the relative difference in total reward gained from starting in state $s_0 = s$ as opposed to $R_\pi$.

Under these definitions, $V^\pi$ is the fixed-point solution to the respective Bellman equations:

$$\textit{Average:} \qquad V^\pi(s) - \mathbb{E}_{s', a|s \sim d_\pi}[V^\pi(s')] = \mathbb{E}_{a|s \sim \pi}[r(s, a) - R_\pi], \qquad (1)$$

$$\textit{Discounted:} \qquad V^\pi(s) - \gamma \mathbb{E}_{s', a|s \sim d_\pi}[V^\pi(s')] = \mathbb{E}_{a|s \sim \pi}[r(s, a)]. \qquad (2)$$

**Importance Sampling**  IS represents a major class of approaches to off-policy estimation, which, in principle, only applies to the finite-horizon reward $R_\pi^T$ when the trajectory is truncated at a finite time step $T < \infty$. IS-based estimators are based on the following change-of-measure equality:

$$R_\pi^T = \mathbb{E}_{\boldsymbol{\tau} \sim p_{\pi_0}}[w_{0:T}(\boldsymbol{\tau})R^T(\boldsymbol{\tau})], \quad \text{with} \quad w_{0:T}(\boldsymbol{\tau}) := \frac{p_\pi(\boldsymbol{\tau}_{0:T})}{p_{\pi_0}(\boldsymbol{\tau}_{0:T})} = \prod_{t=0}^{T} \beta_{\pi/\pi_0}(a_t|s_t), \quad (3)$$

where $\beta_{\pi/\pi_0}(a|s) := \pi(a|s)/\pi_0(a|s)$ is the single-step density ratio of policies $\pi$ and $\pi_0$ evaluated at a particular state-action pair $(s, a)$, and $w_{0:T}$ is the density ratio of the trajectory $\boldsymbol{\tau}$ up to time $T$. Methods based on (3) are called trajectory-wise IS, or weighted IS (WIS) when the weights are self-normalized [22, 30]. It is possible to improve trajectory-wise IS with the so called step-wise, or per-decision, IS/WIS, which uses weight $w_{0:t}$ for reward $r_t$ at time $t$, yielding smaller variance [30]. More details about these estimators are given in Appendix A.

**The Curse of Horizon**  The importance weight $w_{0:T}$ is a product of $T$ density ratios, whose variance can grow exponentially with $T$. Thus, IS-based estimators have not been widely successful in long-horizon problems, let alone infinite-horizon ones where $w_{0:\infty}$ may not even be well-defined. While WIS estimators often have reduced variance, the exponential dependence on horizon is unavoidable in general. We call this phenomenon in IS/WIS-based estimators the *curse of horizon*.

Not all hope is lost, however. To see this, consider an MDP with $n$ states and 2 actions, where states are arranged on a circle (see figure on the right). The two actions deterministically move the agent from the current state to the neighboring state counterclockwise and clockwise, respectively. Suppose we are given two policies with opposite effects: the behavior policy $\pi_0$ moves the agent clockwise with probability $\rho$, and the target policy $\pi$ moves the agent counterclockwise with probability $\rho$, for some constant $\rho \in (0, 1)$. As shown in Appendix B, IS and WIS estimators suffer from exponentially large variance when estimating the average reward of $\pi$. However, a keen reader will realize that the two policies are symmetric, and thus their stationary state visitation distributions are identical. As we show in the sequel, this allows us to estimate the expected reward using a much more efficient importance sampling, whose importance weight equals the single-step density ratio $\beta_{\pi/\pi_0}(a_t|s_t)$, instead of the cumulative product weight $w_{0:T}$ in (3), allowing us to significantly reduce the variance. Such an observation inspired the approach developed in this paper.

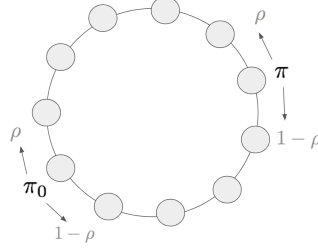

## 3  Off-Policy Estimation via Stationary State Density Ratio Estimation

As shown in the example above, significant decrease in estimation variance is possible when we apply importance weighting on the state space, rather than the trajectory space. It eliminates the dependency on the trajectory length and is much more suited for long- or infinite-horizon problems. To realize this, we need to introduce an alternative representation of the expected reward. Denote by $d_{\pi,t}(\cdot)$ the distribution of state $s_t$ when we execute policy $\pi$ starting from an initial state $s_0$ drawn from an initial distribution $d_0(\cdot)$. We define the average visitation distribution to be

$$d_\pi(s) = \lim_{T \to \infty} \left( \sum_{t=0}^{T} \gamma^t d_{\pi,t}(s) \right) / \left( \sum_{t=0}^{T} \gamma^t \right). \quad (4)$$

We always assume the limit $T \to \infty$ exists in this work. When $\gamma \in (0, 1)$ in the discounted case, $d_\pi$ is a discounted average of $d_{\pi,t}$, that is, $d_\pi(s) = (1 - \gamma) \sum_{t=0}^{\infty} \gamma^t d_{\pi,t}(s)$ ; when $\gamma = 1$ in the average reward case, $d_\pi$ is the stationary distribution of $s_t$ as $t \to \infty$ under policy $\pi$, that is, $d_\pi(s) = \lim_{T \to \infty} \frac{1}{T+1} \sum_{t=0}^{T} d_{\pi,t}(s) = \lim_{t \to \infty} d_{\pi,t}(s)$.

Following Definition 4, it can be verified that $R_\pi$ can be expressed alternatively as

$$R_\pi = \sum_{s,a} d_\pi(s)\pi(a|s)r(s, a) = \mathbb{E}_{(s,a) \sim d_\pi}[r(s, a)], \quad (5)$$

where, abusing notation slightly, we use $(s, a) \sim d_\pi$ to denote draws from distribution $d_\pi(s, a) := d_\pi(s)\pi(a|s)$. Our idea is to construct an IS estimator based on (5), where the importance ratio is

computed on state-action pairs rather than on trajectories:

$$R_\pi = \mathbb{E}_{(s,a)\sim d_{\pi_0}}\left[w_{\pi/\pi_0}(s)\beta_{\pi/\pi_0}(a,s)r(s,a)\right], \tag{6}$$

where $\beta_{\pi/\pi_0}(a,s) = \pi(a|s)/\pi_0(a|s)$ and $w_{\pi/\pi_0}(s) := d_\pi(s)/d_{\pi_0}(s)$ is the density ratio of the visitation distributions $d_\pi$ and $d_{\pi_0}$; here, $w_{\pi/\pi_0}(s)$ is not known directly but can be estimated, as shown later. Eq 5 allows us to construct a (weighted-)IS estimator by approximating $\mathbb{E}_{(s,a)\sim d_{\pi_0}}[\cdot]$ with data $\{s_t^i, a_t^i, r_t^i\}_{i=1}^m$ obtained when running policy $\pi_0$,

$$\hat{R}_\pi = \sum_{i=1}^m \sum_{t=0}^T w_t^i r_t^i, \qquad \text{where} \qquad w_t^i := \frac{\gamma^t w_{\pi/\pi_0}(s_t^i)\beta_{\pi/\pi_0}(a_t^i|s_t^i)}{\sum_{t',i'}\gamma^{t'}w_{\pi/\pi_0}(s_{t'}^{i'})\beta_{\pi/\pi_0}(a_{t'}^{i'}|s_{t'}^{i'})}. \tag{7}$$

This IS estimator works in the space of $(s,a)$, instead of trajectoris $\tau = \{s_t, a_t\}_{t=0}^T$, leading to a potentially significant variance reduction. Returning to the example in Section 2 (see also Appendix B), since the two policies are symmetric and lead to the same stationary distributions, that is, $w_{\pi/\pi_0}(s) = 1$, the importance weight in (6) is simply $\pi(a|s)/\pi_0(a|s)$, independent of the trajectory length. This avoids the excessive variance in long horizon problems. In Appendix A, we provide a further discussion, showing that our estimator can be viewed as a type of *Rao-Backwellization* of the trajectory-wise and step-wise estimators.

## 3.1 Average Reward Case

The key technical challenge remaining is estimating the density ratio $w_{\pi/\pi_0}(s)$, which we address in this section. For simplifying the presentation, we start with estimating $d_\pi(s)$ for the average reward case and discuss the discounted case in Section 3.2.

Let $\boldsymbol{T}_\pi(s'|s) := \sum_a \boldsymbol{T}(s'|s,a)\pi(a|s)$ be the transition probability from $s$ to $s'$ following policy $\pi$. In the average reward case, $d_\pi$ equals the stationary distribution of $\boldsymbol{T}_\pi$, satisfying

$$d_\pi(s') = \sum_s \boldsymbol{T}_\pi(s'|s)d_\pi(s), \quad \forall s'. \tag{8}$$

Assume the Markov chain of $\boldsymbol{T}_\pi$ is finite state and ergodic, $d_\pi$ is also the unique distribution that satisfies (8). This simple fact can be leveraged to derive the following key property of $w_{\pi/\pi_0}(s)$.

**Theorem 1.** *In the average reward case ($\gamma = 1$), assume $d_\pi$ is the unique invariant distribution of $\boldsymbol{T}_\pi$ and $d_{\pi_0}(s) > 0$, $\forall s$. Then a function $w(s)$ equals $w_{\pi/\pi_0}(s) := d_\pi(s)/d_{\pi_0}(s)$ (up to a constant factor) if and only if it satisfies*

$$\begin{aligned} &\mathbb{E}_{(s,a)|s'\sim d_{\pi_0}}[\Delta(w;s,a,s') \mid s'] = 0, \quad \forall s', \\ &\text{with} \quad \Delta(w;s,a,s') := w(s)\beta_{\pi/\pi_0}(a|s) - w(s'), \end{aligned} \tag{9}$$

*where $\beta_{\pi/\pi_0}(a|s) = \pi(a|s)/\pi_0(a|s)$ and $(s,a)|s' \sim d_{\pi_0}$ denote the conditional distribution $d_{\pi_0}(s,a|s')$ related to joint distribution $d_{\pi_0}(s,a,s') := d_{\pi_0}(s)\pi_0(a|s)\boldsymbol{T}(s'|s,a)$. Note that this is a* time-reserved *conditional probability, since it is the conditional distribution of $(s,a)$ given that their next state is $s'$ following policy $\pi_0$.*

Because the conditional distribution is time reversed, it is difficult to directly estimate the conditional expectation $\mathbb{E}_{(s,a)|s'}[\cdot]$ for a given $s'$. This is because we usually can observe only a single data point from $d_{\pi_0}(s,a|s')$ of a fixed $s'$, given that it is difficult to see by chance two different $(s,a)$ pairs transit to the same $s'$. This problem can be addressed by introducing a discriminator function and constructing a mini-max loss function. Specifically, multiplying (9) with a function $f(s')$ and averaging under $s' \sim d_{\pi_0}$ gives

$$\begin{aligned} L(w,f) &:= \mathbb{E}_{(s,a,s')\sim d_{\pi_0}}\left[\Delta(w;s,a,s')f(s')\right] \\ &= \mathbb{E}_{(s,a,s')\sim d_{\pi_0}}\left[\left(w(s)\beta_{\pi/\pi_0}(a|s) - w(s')\right)f(s')\right]. \end{aligned} \tag{10}$$

Following Theorem 1, we have $w \propto w_{\pi/\pi_0}$ if and only if $L(w,f) = 0$ for any function $f$. This motivates us to estimate $w_{\pi/\pi_0}$ with a mini-max problem:

$$\min_w \left\{D(w) := \max_{f\in\mathcal{F}} L\left(w/z_w, \, f\right)^2\right\}, \tag{11}$$

where $\mathcal{F}$ is a set of discriminator functions and $z_w := \mathbb{E}_{s\sim d_{\pi_0}}[w(s)]$ normalizes $w$ to avoid the trivial solution $w \equiv 0$. We shall assume $\mathcal{F}$ to be rich enough following the conditions to be discussed in Section 3.3. A promising choice of a rich function class is neural networks, for which the mini-max problem (11) can be solved numerically in a fashion similar to generative adversarial networks (GANs) [8]. Alternatively, we can take $\mathcal{F}$ to be a ball of a reproducing kernel Hilbert space (RKHS), which enables a closed form representation of $D(w)$ as we show in the following.

**Theorem 2.** *Assume $\mathcal{H}$ is a RKHS of functions $f(s)$ with a positive definite kernel $k(s, \bar{s})$, and define $\mathcal{F} := \{f \in \mathcal{H}\colon ||f||_{\mathcal{H}} \leq 1\}$ to be the unit ball of $\mathcal{H}$. We have*

$$\max_{f\in\mathcal{F}} L(w, f)^2 = \mathbb{E}_{d_{\pi_0}}\left[\Delta(w;\ s, a, s')\Delta(w;\ \bar{s}, \bar{a}, \bar{s}')k(s', \bar{s}')\right], \qquad (12)$$

*where $(s, a, s')$ and $(\bar{s}, \bar{a}, \bar{s}')$ are independent transition pairs obtained when running policy $\pi_0$, and $\Delta(w; s, a, s')$ is defined in (10). See Appendix C for more background on RKHS.*

In practice, we approximate the expectation in (12) using empirical distribution of the transition pairs, yielding consistent estimates following standard results on V-statistics [33].

## 3.2 Discounted Reward Case

We now discuss the extension to the discount case of $\gamma \in (0, 1)$. Similar to the average reward case, we start with a recursive equation that characterizes $d_\pi(s)$ in the discounted case.

**Lemma 3.** *Following the definition of $d_\pi$ in (4), for any $\gamma \in (0, 1]$, we have*

$$\gamma \sum_s \boldsymbol{T}_\pi(s'|s)d_\pi(s) - d_\pi(s') + (1-\gamma)d_0(s') = 0, \quad \forall s'. \qquad (13)$$

*Denote by $(s, a, s') \sim d_\pi$ draws from $d_\pi(s)\pi(a|s)\boldsymbol{T}(s'|s, a)$. For any function $f$, we have*

$$\mathbb{E}_{(s,a,s')\sim d_\pi}[\gamma f(s') - f(s)] + (1-\gamma)\mathbb{E}_{s\sim d_0}[f(s)] = 0. \qquad (14)$$

One may view $d_\pi$ as the invariant distribution of an *induced* Markov chain with transition probability of $(1 - \gamma)d_0(s') + \gamma\boldsymbol{T}_\pi(s'|s)$, which follows $\boldsymbol{T}_\pi$ with probability $\gamma$, and restarts from initial distribution $d_0(s')$ with probability $1 - \gamma$. We can show that $d_\pi$ exists and is unique under mild conditions [31].

**Theorem 4.** *Assume $d_\pi$ is the unique solution of (13), and $d_{\pi_0}(s) > 0, \forall s$. Define*

$$L(w, f) = \gamma\mathbb{E}_{(s,a,s')\sim d_{\pi_0}}[\Delta(w; s, a, s')f(s')] + (1-\gamma)\mathbb{E}_{s\sim d_0}[(1 - w(s))f(s)]. \qquad (15)$$

*Assume $0 < \gamma < 1$, then $w(s) = w_{\pi/\pi_0}(s)$ if and only if $L(w, f) = 0$ for any test function $f$.*

When $\gamma = 1$, the definition in (15) reduces to the average reward case in (10). A subtle difference is that $L(w, f) = 0$ only ensures $w \propto w_{\pi/\pi_0}$ when $\gamma = 1$, while $w = w_{\pi/\pi_0}$ when $\gamma \in (0, 1)$. This is because the additional term $\mathbb{E}_{s\sim d_0}[(1 - w(s))f(s)]$ in (15) forces $w$ to be normalized properly. In practice, however, we still find it works better to pre-normalize $w$ to $\tilde{w} = w/\mathbb{E}_{d_{\pi_0}}[w]$, and optimize the objective $L(\tilde{w},\ f)$.

## 3.3 Further Theoretical Analysis

In this section, we develop further theoretical understanding on the loss function $L(w, f)$. Lemma 5 below reveals an interesting connection between $L(w, f)$ and the Bellman equation, allowing us to bound the estimation error of density ratio and expected reward with the mini-max loss when the discriminator space $\mathcal{F}$ is chosen properly (Theorems 6 and 7). The results in this section apply to both discounted and average reward cases.

**Lemma 5.** *Given $L(w, f)$ in (15), and assuming $\mathbb{E}_{d_{\pi_0}}[w] = 1$ in the average reward case, we have*

$$L(w, f) = \mathbb{E}_{s\sim d_{\pi_0}}[(w_{\pi/\pi_0}(s) - w(s))\Pi f(s)], \qquad (16)$$

$$where \qquad \Pi f(s) := f(s) - \gamma\mathbb{E}_{(s',a)|s\sim d_\pi}[f(s')]. \qquad (17)$$

*Note that $\Pi f$ equals the left hand side of the Bellman equations (1) and (2), when $f = V^\pi$.*

Lemma 5 represents $L(w, f)$ as an inner product between $w_{\pi/\pi_0} - w$ and $\Pi f$ (under base measure $d_{\pi_0}$). This provides an alternative proof of Theorem 4, since $L(w, f) = 0$, $\forall f \in \mathcal{F}$ implies that $w_{\pi/\pi_0} - w$ is orthogonal with all $\Pi f$ and hence $w_{\pi/\pi_0} = w$ when $\{\Pi f \colon f \in \mathcal{F}\}$ is sufficiently rich.

In order to make $(w_{\pi/\pi_0} - w)$ orthogonal to a given function $g$, it requires "reversing" operator $\Pi$: finding a function $f_g$ which solves $g = \Pi f_g$ for given $g$. Observing that $g = \Pi f_g$ can be viewed as a Bellman equation (Eqs. (1)–(2)) when taking $g$ and $f_g$ to be the reward and value functions, respectively, we can derive an explicit representation of $f_g$ (Lemma 10 in Appendix). This allows one to gain insights into what discriminator set $\mathcal{F}$ would be a good choice, so that minimizing $\max_{f \in \mathcal{F}} L(w, f)$ yields good estimation with desirable properties. In the following, by taking $g(s) \propto \pm \mathbf{1}(s = \tilde{s})$, $\forall \tilde{s}$, we can characterize the conditions on $\mathcal{F}$ under which the mini-max loss upper bounds the estimation error of $w_{\pi/\pi_0}$ or $d_\pi$.

**Theorem 6.** *Let $\boldsymbol{T}_\pi^t(s'|s)$ be the t-step transition probability of $\boldsymbol{T}_\pi(s'|s)$. For $\forall \tilde{s} \in \mathcal{S}$, define*

$$f_{\tilde{s}}(s) = \begin{cases} \sum_{t=0}^{\infty} \gamma^t \boldsymbol{T}_\pi^t(\tilde{s}|s) & \text{when } 0 < \gamma < 1, \\ \sum_{t=0}^{\infty} (\boldsymbol{T}_\pi^t(\tilde{s}|s) - d_\pi(\tilde{s})) & \text{when } \gamma = 1, \end{cases} \tag{18}$$

*Assume Lemma 5 holds. We have*

$$\max_{f \in \mathcal{F}} L(w, f) \geq \|d_\pi(s) - w(s) d_{\pi_0}(s)\|_\infty, \qquad \text{if} \quad \{\pm f_{\tilde{s}} \colon \forall \tilde{s} \in \mathcal{S}\} \subseteq \mathcal{F},$$

$$\max_{f \in \mathcal{F}} L(w, f) \geq \|w_{\pi/\pi_0} - w\|_\infty, \qquad \text{if} \quad \{\pm f_{\tilde{s}}/d_{\pi_0}(\tilde{s}) \colon \forall \tilde{s} \in \mathcal{S}\} \subseteq \mathcal{F}.$$

Since our main goal is to estimate the expected total reward $R_\pi$ instead of the density ratio $w_{\pi/\pi_0}$, it is of interest to select $\mathcal{F}$ to directly bound the estimation error of the total reward. Interestingly, this can be achieved once $\mathcal{F}$ includes the true value function $V^\pi$.

**Theorem 7.** *Define $R_\pi[w]$ to be the reward estimate using estimated density ratio $w(s)$ (which may not equal the true ratio $w_{\pi/\pi_0}$) and infinite number of trajectories from $d_{\pi_0}$, that is,*

$$R_\pi[w] := \mathbb{E}_{(s,a,s') \sim d_{\pi_0}} [w(s) \beta_{\pi/\pi_0}(a|s) r(s, a)].$$

*Assume $w$ is properly normalized such that $\mathbb{E}_{s \sim d_{\pi_0}}[w(s)] = 1$, we have $L(w, V^\pi) = R_\pi - R_\pi[w]$. Therefore, if $\pm V^\pi \in \mathcal{F}$, we have $|R_\pi[w] - R_\pi| \leq \max_{f \in \mathcal{F}} L(w, f)$.*

## 4 Related Work

Our off-policy setting is related to, but different from, off-policy value-function learning [30, 29, 37, 12, 25, 21]. Our goal is to estimate a single scalar that *summarizes* the quality of a policy (a.k.a. off-policy value estimation as called by some authors [20]). However, our idea can be extended to estimating value functions as well, by using estimated density ratios to weight observed transitions (c.f., the distribution $\mu$ in LSTDQ [16]). We leave this as future work.

IS-based off-policy value estimation has seen a lot of interest recently for short-horizon problems, including contextual bandits [26, 13, 7, 42], and achieved many empirical successes [7, 34]. When extended to long-horizon problems, it faces an exponential blowup of variance, and variance-reduction techniques are used to improve the estimator [14, 39, 10, 42]. However, it can be proved that in the worst case, the mean squared error of *any* estimator has to depend exponentially on the horizon [20, 10]. Fortunately, many problems encountered in practical applications may present structures that enable more efficient off-policy estimation, as tackled by the present paper. An interesting open direction is to characterize theoretical conditions that can ensure tractable estimation for long horizon problems.

Few prior work directly target *infinite*-horizon problems. There exists approaches that use simulated samples to estimate stationary state distributions [1, Chapter IV]. However, they need a reliable model to draw such simulations, a requirement that is not satisfied in many real-world applications. To the best of our knowledge, the recently developed COP-TD algorithm [11] is the only work that attempts to estimate $w_{\pi/\pi_0}$ as an intermediate step of estimating the value function of a target policy $\pi$. They take a stochastic-approximation approach and show asymptotic consistence. However, extending their approach to continuous state/action spaces appears challenging.

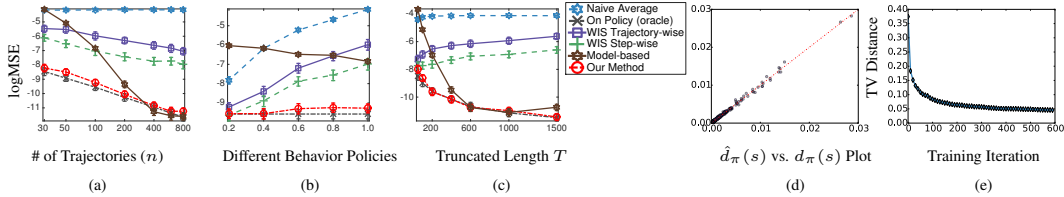

Figure 1: Results on Taxi environment with average reward ($\gamma = 1$). (a)-(b) show the performance of various methods as the number of trajectory (a) and the difference between behavior and target policies (b) vary. (c) shows the change of truncated length $T$. (d) shows that scatter plot of pairs $(\hat{d}_\pi(s), d_\pi(s))$, $\forall s$. The diagonal lines means exact estimation. (e) shows the weighted total variation distance between $\hat{d}_\pi := \hat{w}d_{\pi_0}$ and $d_\pi$ along the training iteration of the ratio estimator $\hat{w}$. The number of trajectory is fixed to be 100 in (b,c,d). The potential behavior policy $\pi_+$ (the right most points in (b)) is used in (a,c,d,e).

Finally, there is a comprehensive literature of two-sample density ratio estimation [e.g., 27, 35], which estimates the density ratio of two distributions from pairs of their samples. Our problem setting is different in that we only have data from $d_{\pi_0}$, but not from $d_\pi$; this makes the traditional density ratio estimators inapplicable to our problem. Our method is made possible by taking the special temporal structure of MDP into consideration.

## 5    Experiment

In this section, we conduct experiments on different environmental settings to compare our method with existing off-policy evaluation methods. We compare with the standard trajectory-wise and step-wise IS and WIS methods. We do not report the results of unnormalized IS because they are generally significantly worse than WIS methods [30, 22]. In all the cases, we also compare with an *on-policy oracle* and a *naive averaging* baseline, which estimates the reward using direct averaging over the trajectories generated by the target policy and behavior policy, respectively. For problems with discrete action and state spaces, we also compare with a standard model-based method, which estimates the transition and reward model and then calculates expected reward explicitly using the model up to the desired truncation length. When applying our method on problems with finite and discrete state space, we optimize $w$ and $f$ in the space of all possible functions (corresponding to using a delta kernel in terms of RKHS). For continuous state space, we assume $w$ is a standard feed-forward neural network, and $\mathcal{F}$ is a RKHS with a standard Gaussian RBF kernel whose bandwidth equals the median of the pairwise distances between the observed data points.

Because we cannot simulate truly infinite steps in practice, we use the behavior policy to generate trajectories of length $T$, and evaluate the algorithms based on the mean square error (MSE) w.r.t. the $T$-step rewards of a large number of trajectories of length $T$ from the target policy. We expect that our method gets better as $T$ increases, since it is designed for infinite horizon problems, while the IS/WIS methods receive large variance and deteriorate as $T$ increases.

**Taxi Environment**    Taxi [6] is a 2D grid world simulating taxi movement along the grids. A taxi moves North, East, South, West or attends to pick up or drop off a passenger. It receives a reward of 20 when it successfully picks up a passenger or drops her off at the right place, and otherwise a reward of -1 every time step. The original taxi environment would stop when the taxi successfully picks up a passenger and drops her off at the right place. We modify the environment to make it infinite horizon, by allowing passengers to randomly appear and disappear at every corner of the map at each time step. We use a grid size of $5 \times 5$, which yields 2000 states in total ($25 \times 2^4 \times 5$, corresponding to 25 taxi locations, $2^4$ passenger appearance status and 5 taxi status (empty or with one of 4 destinations)).

To construct target and behavior policies for testing our algorithm, we set our target policy to be the final policy $\pi_*$ after running Q-learning for 1000 iterations, and set another policy $\pi_+$ after 950 iterations. The behavior policy is $\pi = (1 - \alpha)\pi_* + \alpha\pi_+$, where $\alpha$ is a mixing ratio that can be varied.

**Results in Taxi Environment**    Figure 1(a)–(b) show results with average reward. We can see our method performs almost as well as the on-policy oracle, outperforming all the other methods. To evaluate the approximation error of the estimated density ratio $\hat{w}$, we plot in Figure 1(c) the weighted

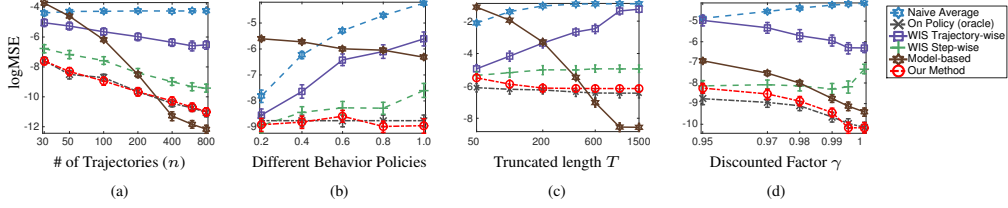

Figure 2: Results on Taxi with discounted reward ($0 < \gamma < 1$), as we vary the number of trajectory $n$ (a), the difference between target and behavior policies (b), the truncated length $T$(c), the discount factor $\gamma$ (d). The default values of the parameters, unless it is varying, are $\gamma = 0.99$, $n = 200$, $T = 400$. The potential behavior policy $\pi_+$ (the right most points in (b)) is used in (a,c,d).

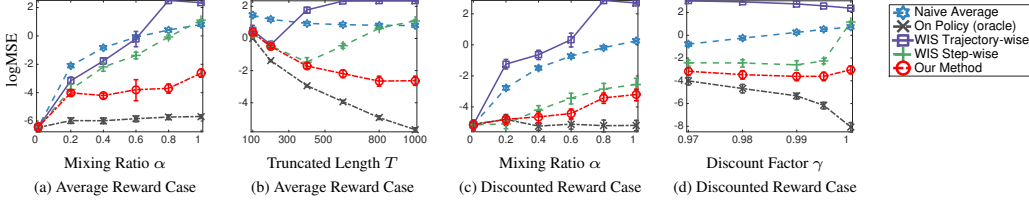

Figure 3: Results on Pendulum. (a)-(b) show the results in the average reward case when we vary the mixing ratio $\alpha$ in the behavior policies and the truncated length $T$, respectively. (c)-(d) show the results of the discounted reward case when we vary mixing ratio $\alpha$ in the behavior policies and discount factor $\gamma$, respectively. The default parameters are $n = 150$, $T = 1000$, $\gamma = 0.99$, $\alpha = 1$.

total variation distance between $\hat{d}_\pi = \hat{w} d_{\pi_0}$ with the true $d_\pi$ with TV distance as we optimize the loss function. Figure 1(d) shows scatter plot of $\{(\hat{d}_\pi(s), d_\pi(s)) : \forall s \in \mathcal{S}\}$ at convergence, indicating our method correctly estimates the true density ratio over the state space.

Figure 2 shows similar results for discounted reward. From Figure 2(c) and (d), we can see that typical IS methods deteriorate as the trajectory length $T$ and discount factor $\gamma$ increase, respectively, which is expected since their variance grows exponentially with $T$. In contrast, our density ratio method performs better as trajectory length $T$ increases, and is robust as $\gamma$ increases.

**Pendulum Environment**   The Taxi environment features discrete action and state spaces. We now test Pendulum, which has a continuous state space of $\mathbb{R}^3$ and action space of $[-2, 2]$. In this environment, we want to control the pendulum to make it stand up as long as possible (for the average case), or as fast as possible (for small discounted case). The policy is taken to be a truncated Gaussian whose mean is a neural network of the states and variance a constant.

We train a near-optimal policy $\pi_*$ using REINFORCE and set it to be the target policy. The behavior policy is set to be $\pi = (1 - \alpha)\pi_* + \alpha\pi_+$, where $\alpha$ is a mixing ratio, and $\pi_+$ is another policy from REINFORCE when it has not converged. Our results are shown in Figure 3, where we again find that our method generally outperforms the standard trajectory-wise and step-wise WIS, and works favorably in long-horizon problems (Figure 3(b)).

**SUMO Traffic Simulator**   SUMO [15] is an open source traffic simulator; see Figure 4(a) for an illustration. We consider the task of reducing traffic congestion by modelling traffic light control as a reinforcement learning problem [41]. We use TraCI, a built-in "Traffic Control Interface", to interact

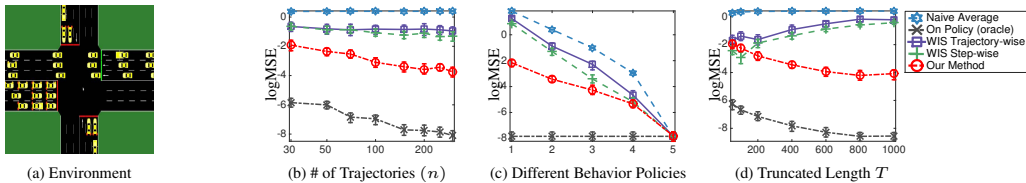

Figure 4: Results on SUMO (a) with average reward, as we vary the number of trajectories (b), choose different behavior policies (c), and truncated size (d). When being fixed, the default parameters are $n = 250$, $T = 400$. The behavior policy in (c) with x-tick 2 is used in (b) and (d).

with the SUMO simulator. Full details of our environmental settings can be found in Appendix E. Our results are shown in Figure 4, where we again find that our method is consistently better than standard IS methods.

## 6 Conclusions

We study the off-policy estimation problem in infinite-horizon problems and develop a new algorithm based on direct estimation of the stationary state density ratio between the target and behavior policies. Our mini-max objective function enjoys nice theoretical properties and yields an intriguing connection with Bellman equations that is worth further investigation. Future directions include scaling our method to larger scale problems and extending it to estimate value functions and leverage off-policy data in policy optimization.

## Acknowledgement

This work is supported in part by NSF CRII 1830161. We would like to acknowledge Google Cloud for their support.

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
