[Supplementary Material]

# A  Several Variants of IS- and WIS-based Estimators

Denote by $\gamma_t = \gamma^t / \sum_{t=0}^{T} \gamma^t$ for notation simplicity. Define

$$w_{0:T}(\boldsymbol{\tau}) := \prod_{t=0}^{T} \frac{\pi(a_t|s_t)}{\pi_0(a_t|s_t)}.$$

Then we have the following two key formulas, which derive the trajectory-wise, and step-wise importance sampling (IS) estimators, respectively.

$$R_\pi^T = \mathbb{E}_{\boldsymbol{\tau} \sim p_{\pi_0}} \left[ \sum_{t=0}^{T} w_{0:T}(\boldsymbol{\tau}) \gamma_t r_t \right] \qquad \textit{(Trajectory-wise)} \qquad (19)$$

$$= \mathbb{E}_{\boldsymbol{\tau} \sim p_{\pi_0}} \left[ \sum_{t=0}^{T} w_{0:t}(\boldsymbol{\tau}) \gamma_t r_t \right] \qquad \textit{(Step-wise)} \qquad (20)$$

where the only difference of (19) and (20) is that (20) replaces the $w_{0:T}$ in (19) with $w_{0:t}$, yielding smaller variance without changing the expectation. This is made possible because $w_{0:t} = \mathbb{E}_{\boldsymbol{\tau} \sim p_{\pi_0}}[w_{0:T}(\boldsymbol{\tau}) \mid \boldsymbol{\tau}_{0:t}]$. Therefore, step-wise estimator can be viewed as Rao-backwellizing each term $w_{0:T}(\boldsymbol{\tau}) \gamma_t r_t$ in (19) by conditioning on $\boldsymbol{\tau}_{0:t}$.

Given a set of $m$ observed trajectories $\boldsymbol{\tau}^i = \{s_t^i, a_t^i, r_t^i\}_{t=0}^{T}$, $\forall i = 1, \dots, m$, drawn from $p_{\pi_0}$. The trajectory-wise and step-wise estimators are

$$\textit{Trajectory-wise:} \quad \hat{R}_\pi^T = \frac{1}{Z_T} \sum_{t=0}^{T} \sum_{i=1}^{m} \gamma_t w_{0:T}^i r_t^i, \quad \textit{Step-wise:} \quad \hat{R}_\pi^T = \sum_{t=0}^{T} \sum_{i=1}^{m} \frac{1}{Z_t} \gamma_t w_{0:t}^i r_t^i,$$

where $w_{0:t}^i = w_{0:t}(\boldsymbol{\tau}^i)$ and $Z_t$ is a normalization constant of the importance weights: when $Z_t = m, \quad \forall t$, the corresponding estimators (called Trajectory-wise IS and Step-wise IS, respectively) provide unbiased estimates of $R_\pi^T$; when $Z_t = \sum_{i=1}^{m} w_{0:t}^i$, the corresponding estimators are weighted (or self-normalized) importance sampling (called Trajectory-wise WIS and Step-wise WIS, respectively), which introduce bias but often have lower variance. It has been shown that the Step-wise WIS often performs the best among all these variants [30, 22].

In comparison, our method can be viewed as a further Rao-backwellization of the step-wise estimators. Define

$$w_{t:t}(a_t, s_t) = \mathbb{E}_{\boldsymbol{\tau} \sim p_{\pi_0}} \left[ w_{0:T}(\boldsymbol{\tau}) \mid (s_t, a_t) \right] = \frac{d_\pi(s_t)}{d_{\pi_0}(s_t)} \frac{\pi(a_t|s_t)}{\pi_0(a_t|s_t)}.$$

Then we have

$$R_\pi^T = \mathbb{E}_{\boldsymbol{\tau} \sim p_{\pi_0}} \left[ \sum_{t=0}^{T} w_{t:t}(a_t, s_t) \gamma_t r_t \right] \qquad \textit{(Our method)}, \qquad (21)$$

where we replace $w_{0:t}$ in (20) with $w_{t:t}$, based on Rao-backwellization conditioning on $(s_t, a_t)$. This gives an empirical estimator:

$$\textit{Our method:} \qquad \hat{R}_\pi^T = \sum_{t=0}^{T} \sum_{i=1}^{m} \frac{1}{Z_t} \gamma_t w_{t:t}^i r_t^i,$$

where $w_{t:t}^i = w_{t:t}(a_t^i, s_t^i)$ and $Z_t = m$ or $Z_t = \sum_{i=1}^{m} w_{t:t}^i$. Comparing this with the trajectory-wise and step-wise estimators, it is easy to expect that it yields smaller variance, when ignoring the estimation error of $w_{t:t}$.

# B  A motivating example

Here we provide an example when $w_{0:T}$ is exponential on the trajectory length $T$, yielding high variance in trajectory-wise and step-wise estimators in long horizon problems, while the variance of our stationary density ratio based importance weight $w_{t:t}$ stays to be a constant as $T$ increases.

The MDP has $n$ states: $\mathcal{S} = \{0, 1, \ldots, n-1\}$, arranged on a circle (see the figure on the right), where $n$ is an odd number. There are two actions, left (L) and right (R). The left action moves the agent from the current state counterclockwise to the next state, and the right action has the opposite (clockwise) effect. The deterministic reward is $0$ if taking action L and $1$ otherwise. In summary, we have for any $s$ and $a$ that

$$\boldsymbol{T}(s'|s, \mathsf{L}) = \mathbb{I}(s' = s - 1 \bmod n)$$
$$\boldsymbol{T}(s'|s, \mathsf{R}) = \mathbb{I}(s' = s + 1 \bmod n)$$
$$r(s, a) = \mathbb{I}(a = \mathsf{R}).$$

Suppose we are given two policies. The behavior policy $\pi_0$ and target policy $\pi$ choose action R with probability $\rho$ and $1 - \rho$, respectively. We focus on the average reward $(\gamma = 1)$ here.

**Claim #1. Stationary density ratio $w_{t:t}$ stays constant as $t \to \infty$.** First, note that the MDP is ergodic under either policy, as $n$ is odd. Since $\pi_0$ and $\pi$ are symmetric, their stationary distributions are identical, that is, $d_\pi(s)/d_{\pi_0}(s) = 1$. In fact, both $d_\pi = d_{\pi_0}$ are uniform over $\mathcal{S}$. Therefore,

$$w_{t:t}(s, \mathsf{R}) = \frac{d_\pi(s)\pi(\mathsf{R}|s)}{d_{\pi_0}(s)\pi_0(\mathsf{R}|s)} = \frac{\pi(\mathsf{R}|s)}{\pi_0(\mathsf{R}|s)} = \frac{\rho}{1 - \rho},$$

and similarly $w_{t:t}(s, \mathsf{L}) = (1 - \rho)/\rho$. Both ratios are *independent* of the trajectory length, and have *zero* variance.

**Claim #2. Variance of trajectory-wise IS weight $w_{0:T}$ grows exponentially in $T$.**

**Proposition 8.** *Under the setting above, let $\boldsymbol{\tau} = \{s_t, a_t, r_t\}_{0 \le t \le T}$ be a trajectory drawn from the behavior policy $\pi_0$, we have*

$$\mathrm{var}_{p_{\pi_0}}[w_{0:T}(\boldsymbol{\tau})] = A_\rho^{T+1} - 1,$$
$$\mathrm{var}_{p_{\pi_0}}[w_{0:T}(\boldsymbol{\tau})R^T(\boldsymbol{\tau})] = B_{\rho,T}A_\rho^{T-1} - (1-\rho)^2,$$

*where*

$$A_\rho := \frac{\rho^3 + (1-\rho)^3}{(1-\rho)\rho}, \qquad B_{\rho,T} = \frac{(1-\rho)\rho}{T+1} + \frac{(1-\rho)^4}{\rho}.$$

*Obviously, $A_\rho > 1$ for $\rho \ne 1/2$ and $A_\rho = 1$ for $\rho = 1/2$, and $B_{\rho,T} > 0$ for large enough $T$. Therefore, the variance of both the trajectory-wise importance weights and the corresponding estimator grow exponentially in the order of $A_\rho^T$.*

**Remark** When $\rho = 1/2$, it reduces to the on-policy case of $\pi = \pi_0$, for which we can show that $\mathrm{var}_{p_{\pi_0}}[w_{0:T}(\boldsymbol{\tau})] = 0$ (since $w_{0:T}(\boldsymbol{\tau}) = 1$), and $\mathrm{var}_{p_{\pi_0}}[w_{0:T}(\boldsymbol{\tau})R^T(\boldsymbol{\tau})] = 1/(4(T+1))$.

*Proof.* From the definition of the setting, it is easy to show that

$$R^T(\boldsymbol{\tau}) = \frac{F(\boldsymbol{\tau})}{T+1}, \qquad w_{0:T}(\boldsymbol{\tau}) = \prod_{t=0}^{T} \frac{\pi(a_t|s_t)}{\pi_0(a_t|s_t)} = \left(\frac{1-\rho}{\rho}\right)^{2F(\boldsymbol{\tau})-(T+1)}$$

where

$$F(\boldsymbol{\tau}) = \sum_{t=0}^{T} \mathbb{I}(a_t = \mathsf{R}).$$

Under policy $\pi_0$, $F(\boldsymbol{\tau})$ follows a Binomial distribution $Binomial(T+1, \rho)$. The first order moments can be easily calculated as follows

$$\mathbb{E}_{\boldsymbol{\tau} \sim p_{\pi_0}}[w_{0:T}(\boldsymbol{\tau})] = 1, \qquad \mathbb{E}_{\boldsymbol{\tau} \sim p_{\pi_0}}[w_{0:T}(\boldsymbol{\tau})R^T(\boldsymbol{\tau})] = \mathbb{E}_{\boldsymbol{\tau} \sim p_\pi}[R^T(\boldsymbol{\tau})] = 1 - \rho.$$

It remains to calculate the second order moments. We achieve this by leveraging the moment-generating function (MGF) of Binomial distribution:

$$\Phi(\lambda) := \mathbb{E}_{\boldsymbol{\tau}\sim p_{\pi_0}}[\exp(\lambda F(\boldsymbol{\tau}))] = (1 - \rho + \rho\exp(\lambda))^{T+1}, \quad \forall\lambda \in \mathbb{R}. \tag{22}$$

It will turn out be useful to consider the derivatives of $\Phi(\lambda)$:

$$\Phi'(\lambda) = \mathbb{E}_{\boldsymbol{\tau}\sim p_{\pi_0}}[\exp(\lambda F(\boldsymbol{\tau}))F(\boldsymbol{\tau})]$$
$$= (T+1)(1 - \rho + \rho\exp(\lambda))^T \rho\exp(\lambda),$$

and

$$\Phi''(\lambda) = \mathbb{E}_{\boldsymbol{\tau}\sim p_{\pi_0}}[\exp(\lambda F(\boldsymbol{\tau}))F(\boldsymbol{\tau})^2]$$
$$= (T+1)(1 - \rho + \rho\exp(\lambda))^{T-1}(1 - \rho + (T+1)\rho\exp(\lambda))\rho\exp(\lambda). \tag{23}$$

For convenience, define $C = (1 - \rho)/\rho$, and we have

$$\mathbb{E}_{\boldsymbol{\tau}\sim p_{\pi_0}}[w_{0:T}(\boldsymbol{\tau})^2] = \mathbb{E}_{\boldsymbol{\tau}\sim p_{\pi_0}}[(C^{2F(\boldsymbol{\tau})-(T+1)})^2]$$
$$= \Phi(4\log C)\cdot C^{-2(T+1)}$$
$$= \left[(1 - \rho + \rho C^4)C^{-2}\right]^{T+1}$$
$$= A_\rho^{T+1},$$

where we use the fact that $(1 - \rho + \rho C^4)C^{-2} = \frac{\rho^3 + (1-\rho)^3}{(1-\rho)\rho} = A_\rho$. Similarly, we have

$$\mathbb{E}_{\boldsymbol{\tau}\sim p_{\pi_0}}\left[w_{\pi/\pi_0}(\boldsymbol{\tau})^2 R(\boldsymbol{\tau})^2\right]$$
$$= \mathbb{E}_{\boldsymbol{\tau}\sim p_{\pi_0}}\left[C^{4F(\boldsymbol{\tau})-2(T+1)}F(\boldsymbol{\tau})^2\right]/(T+1)^2$$
$$= \Phi''(4\log C)C^{-2(T+1)}/(T+1)^2$$
$$= ((1 - \rho + \rho C^4)C^{-2})^{T-1}(C/(T+1) + \rho C^4)\rho^2$$
$$= B_{\rho,T}A_\rho^{T-1}$$

where we use the fact that $B_{\rho,T} = (C/(T+1) + \rho C^4)\rho^2$. It is then straightforward to calculate the variance from here. $\qquad\square$

**Claim #3. Variance of trajectory-wise WIS weight grows exponentially in $T$.** Although weighted-IS (WIS) often improves over IS estimators by using self-normalized weights, it cannot eliminate the exponential dependence on the trajectory length. Here, we calculate the asymptotic variance of trajectory-wise WIS using delta method [28, Chapter 9].

**Proposition 9.** *Let $\hat{R}_{n,wis}$ be the trajectory-wise WIS estimator of $R_\pi$ based on $n$ copies of independent trajectories drawn from $\pi_0$, we have*

$$\mathbb{E}_{p_{\pi_0}}[(\hat{R}_{n,wis} - R_\pi)^2] = \frac{1}{n}D_{\rho,T}A_\rho^T + o\left(\frac{1}{n}\right),$$

*where $D_{\rho,A} = B_{\rho,T}A_\rho^{-1} - 2(1-\rho)^3/\rho + (1-\rho)^2 A_\rho$, with $A_\rho$ and $B_{\rho,T}$ defined in Proposition 8.*

*Proof.* The asymptotic mean square error (MSE) of a self-normalized importance sampling estimator can be estimated using the delta method [28, Chapter 9]:

$$\mathbb{E}_{p_{\pi_0}}[(\hat{R}_{n,\text{wis}} - R_\pi)^2]$$
$$= \frac{1}{n}\mathbb{E}_{\boldsymbol{\tau}\sim p_{\pi_0}}\left[w_{\pi/\pi_0}(\boldsymbol{\tau})^2(R(\boldsymbol{\tau}) - R_\pi)^2\right] + o\left(\frac{1}{n}\right).$$

Note that

$$\mathbb{E}_{\boldsymbol{\tau}\sim p_{\pi_0}}\left[w_{\pi/\pi_0}(\boldsymbol{\tau})^2(R(\boldsymbol{\tau}) - R_\pi)^2\right]$$
$$= \mathbb{E}_{\boldsymbol{\tau}\sim p_{\pi_0}}\left[w_{\pi/\pi_0}(\boldsymbol{\tau})^2 R(\boldsymbol{\tau})^2\right] - 2R_\pi\mathbb{E}_{\boldsymbol{\tau}\sim p_{\pi_0}}[w_{\pi/\pi_0}(\boldsymbol{\tau})^2 R(\boldsymbol{\tau})] + R_\pi^2\mathbb{E}_{\boldsymbol{\tau}\sim p_{\pi_0}}\left[w_{\pi/\pi_0}(\boldsymbol{\tau})^2\right],$$

where the first and third terms have been calculated in the proof of Proposition 8. We just need to calculate the cross term:

$$\mathbb{E}_{\boldsymbol{\tau}\sim p_{\pi_0}}[w_{\pi/\pi_0}(\boldsymbol{\tau})^2 R(\boldsymbol{\tau})] = \mathbb{E}_{\boldsymbol{\tau}\sim p_{\pi_0}}\left[C^{4F(\boldsymbol{\tau})-2(T+1)}F(\tau)\right]/(T+1)$$
$$= \Phi'(4\log C)C^{-2(T+1)}/(T+1)$$
$$= \left[(1-\rho+\rho C^4)C^{-2}\right]^T \rho C^2$$
$$= (1-\rho)^2/\rho A_\rho^T.$$

Therefore,

$$\mathbb{E}_{\boldsymbol{\tau}\sim p_{\pi_0}}\left[w_{\pi/\pi_0}(\boldsymbol{\tau})^2(R(\boldsymbol{\tau})-R_\pi)^2\right] = B_{\rho,T}A_\rho^{T-1} - 2R_\pi(1-\rho)^2/\rho A_\rho^T + R_\pi^2 A_\rho^{T+1}$$
$$= D_{\rho,T}A_\rho^T,$$

where

$$D_{\rho,T} := B_{\rho,T}A_\rho^{-1} - 2R_\pi(1-\rho)^2/\rho + R_\pi^2 A_\rho$$
$$= B_{\rho,T}A_\rho^{-1} - 2(1-\rho)^3/\rho + (1-\rho)^2 A_\rho.$$

We used $R_\pi = 1 - \rho$ here.

## C   Proofs

**Reproducing Kernel Hilbert Space (RKHS)**   We start with a brief, informal introduction of RKHS. A symmetric function $k(s, s')$ is called positive definite if all matrices of form $[k(s_i, s_j)]_{ij}$ are positive definite for any $\{s_i\} \subseteq \mathcal{S}$. Related to every positive definite kernel $k(s, s')$ is an unique RKHS $\mathcal{H}$ which is the closure of functions of form $f(s) = \sum_i a_i k(s, s_i), \forall a_i \in \mathbb{R}, s_i \in \mathcal{S}$, equipped with a norm and inner product defined as

$$\langle f, g\rangle_{\mathcal{H}} = \sum_{ij} a_i b_j k(s_i, s_j), \qquad\qquad \|f\|_{\mathcal{H}}^2 = \sum_{ij} a_i a_j k(s_i, s_j),$$

where we assume $g(x) = \sum_i b_i k(s, s_i)$. A simple yet important fact that our proof will leverage is that

$$\|f\|_{\mathcal{H}} = \max_{g\in\mathcal{F}}\langle f, g\rangle_{\mathcal{H}}, \qquad \text{where} \qquad \mathcal{F} = \{g \in \mathcal{H}\colon \|g\|_{\mathcal{H}} \le 1\}.$$

A key property of RKHS is the so called reproducing property, which says

$$f(s) = \langle f(\cdot),\ k(s,\cdot)\rangle_{\mathcal{H}}, \qquad \text{and hence} \qquad k(s, s') = \langle k(s,\cdot),\ k(s',\cdot)\rangle_{\mathcal{H}}.$$

In our proof, we will consider functions of form $f(s) = \mathbb{E}_{s'\sim d}[w(s')k(s, s')]$ for some function $w$ and distribution $d$, for which one can show that

$$\max_{g\in\mathcal{F}}\langle f,\ g\rangle_{\mathcal{H}} = \|f\|_{\mathcal{H}} = \mathbb{E}_{s,s'\sim d}[w(s)w(s')k(s, s')]^{1/2};$$

this can be proved using the reproducing property as follows

$$\|f\|_{\mathcal{H}}^2 = \langle f, f\rangle_{\mathcal{H}} = \langle \mathbb{E}_{s\sim d}[w(s)k(\cdot, s)],\ \mathbb{E}_{s'\sim d}[w(s')k(\cdot, s')]\rangle_{\mathcal{H}}$$
$$= \mathbb{E}_{s,s'\sim d}[w(s)w(s')\langle k(\cdot, s),\ k(\cdot, s')\rangle_{\mathcal{H}}]$$
$$= \mathbb{E}_{s,s'\sim d}[w(s)w(s')k(s, s')].$$

For more introduction to RKHS, see [32, 3, 24], to name only a few.

*Proof of Theorem 1.* Note that $d_{\pi_0}(s, a|s') = \frac{d_{\pi_0}(s)\pi_0(a|s)\boldsymbol{T}(s'|s,a)}{d_{\pi_0}(s')}$. Therefore, (9) is equivalent to

$$w(s') = \mathbb{E}_{(s,a)|s'\sim\pi_0}\left[w(s)\frac{\pi(a|s)}{\pi_0(a|s)}\ \middle|\ s'\right] = \sum_{s,a}\frac{d_{\pi_0}(s)\pi_0(a|s)\boldsymbol{T}(s'|s,a)}{d_{\pi_0}(s')}w(s)\frac{\pi(a|s)}{\pi_0(a|s)}$$
$$= \frac{1}{d_{\pi_0}(s')}\sum_{s,a}\boldsymbol{T}(s'|s,a)\pi(a|s)d_{\pi_0}(s)w(s), \qquad \forall s'.$$

Denote $g(s) := d_{\pi_0}(s)w(s)$. Since $d_{\pi_0}(s') > 0$ for all $s'$, we find that (9) is equivalent to

$$g(s') = \sum_{s,a} \boldsymbol{T}(s'|s,a)\pi(a|s)g(s), \qquad \forall s'. \tag{24}$$

This implies that $g(s)$ is invariant under Markov transition $\boldsymbol{T}(s'|s,a)\pi(a|s)$. Because $d_\pi(s)$ is the unique stationary distribution under the same Markov transition, (24) holds if and only if $g(s) \propto d_\pi(s)$, or equivalently, $w(s) \propto w_{\pi/\pi_0}(s)$. This completes the proof. $\qquad\square$

*Proof of Theorem 2.* By the reproducing property of RKHS, we have $f(s) = \langle f(\cdot), k(s,\cdot)\rangle_{\mathcal{H}}$. This gives $L(w,f) = \langle f, \phi^*\rangle_{\mathcal{H}}$, where $\phi^*(\cdot) = \mathbb{E}_{\pi_0}[\Delta(w; \bar{s}, \bar{a}, \bar{s}')k(\bar{s}', \cdot)]$. The results then follow by

$$\max_f L(w,f)^2 = \max_{f\in\mathcal{F}}\langle f, \phi^*\rangle_{\mathcal{H}}^2 = \|\phi^*\|_{\mathcal{H}}^2 = \mathbb{E}_{\pi_0}\left[\Delta(w;\ s,a,s')\Delta(w;\ \bar{s},\bar{a},\bar{s}')k(s',\bar{s}')\right].$$

$\square$

*Proof of Lemma 3.* Assume $\gamma \in (0,1)$. The definition in (4) gives $d_\pi(s) = (1-\gamma)\sum_{t=0}^{\infty}\gamma^t d_{\pi,t}(s)$. Therefore,

$$d_\pi(s') = (1-\gamma)\sum_{t=0}^{\infty}\gamma^t d_{\pi,t}(s')$$

$$= (1-\gamma)d_0(s') + (1-\gamma)\sum_{t=1}^{\infty}\gamma^t d_{\pi,t}(s')$$

$$= (1-\gamma)d_0(s') + (1-\gamma)\gamma\sum_{t=0}^{\infty}\gamma^t d_{\pi,t+1}(s')$$

$$= (1-\gamma)d_0(s') + (1-\gamma)\gamma\sum_{t=0}^{\infty}\gamma^t \sum_s \boldsymbol{T}_\pi(s'|s)d_{\pi,t}(s) \qquad // \ d_{\pi,t+1}(s') = \sum_{s,a}\boldsymbol{T}_\pi(s'|s)d_{\pi,t}(s)$$

$$= (1-\gamma)d_0(s') + \gamma\sum_s \boldsymbol{T}_\pi(s'|s)\left((1-\gamma)\sum_{t=0}^{\infty}\gamma^t d_{\pi,t}(s)\right)$$

$$= (1-\gamma)d_0(s') + \gamma\sum_s \boldsymbol{T}_\pi(s'|s)d_\pi(s)$$

$$= (1-\gamma)d_0(s') + \gamma\sum_{s,a}\boldsymbol{T}(s'|s,a)\pi(a|s)d_\pi(s).$$

Multiplying both sides by $f(s')$ and summing over $s'$, we get

$$\sum_{s'}d_\pi(s')f(s') = (1-\gamma)\sum_{s'}d_0(s')f(s') + \gamma\sum_{s,a,s'}\boldsymbol{T}(s'|s,a)\pi(a|s)d_\pi(s)f(s').$$

Recall that $(s,a,s') \sim d_\pi$ denotes sampling from the joint distribution of $d_\pi(s,a,s') = d_\pi(s)\boldsymbol{T}(s',a|s)\pi(a|s)$. Note that under this joint distribution, the marginal distribution of $s'$ is different from $d_\pi(s)$.[1]

The above equation is equivalent to

$$\mathbb{E}_{s'\sim d_\pi}[f(s')] = (1-\gamma)\mathbb{E}_{s'\sim d_0}[f(s')] + \gamma\mathbb{E}_{(s,a,s')\sim d_\pi}[f(s')].$$

For notation, changing the dummy variable $s'$ in $\mathbb{E}_{s'\sim d_\pi}[\cdot]$ and $\mathbb{E}_{s'\sim d_0}[\cdot]$ to $s$ gives

$$\mathbb{E}_{s\sim d_\pi}[f(s)] = (1-\gamma)\mathbb{E}_{s\sim d_0}[f(s)] + \gamma\mathbb{E}_{(s,a,s')\sim d_\pi}[f(s')].$$

Therefore,

$$\mathbb{E}_{(s,a,s')\sim d_\pi}[\gamma f(s') - f(s)] + (1-\gamma)\mathbb{E}_{s\sim d_0}[f(s)] = 0.$$

$\square$

*Proof of Theorem 4.* Define

$$\delta(g,\ s') := \gamma \sum_s \boldsymbol{T}_\pi(s'|s)g(s) - g(s') + (1-\gamma)d_0(s'),$$

where $g$ is any function. Then by assumption, we have $g(s) = d_\pi(s)$ if and only if $\delta(g,\ s') = 0$ for any $s'$. Replacing $d_\pi$ with $d_{\pi_0}$ and $f(s)$ with $w(s)f(s)$ in (14) gives

$$\mathbb{E}_{(s,a,s')\sim d_{\pi_0}}[w(s)f(s) - \gamma w(s')f(s')] = (1-\gamma)\mathbb{E}_{s\sim d_0}[w(s)f(s)].$$

Plugging it into the definition of $L(w,f)$ in (15), we get

$$
\begin{aligned}
L&(w,f) \\
&= \gamma\mathbb{E}_{(s,a,s')\sim d_{\pi_0}}[(\beta_{\pi/\pi_0}(a|s)w(s) - w(s'))f(s')] + (1-\gamma)\mathbb{E}_{s\sim d_0}[(1-w(s))f(s)] \\
&= \gamma\mathbb{E}_{(s,a,s')\sim d_{\pi_0}}[(\beta_{\pi/\pi_0}(a|s)w(s)f(s')] - \mathbb{E}_{s\sim d_{\pi_0}}[w(s)f(s)] + (1-\gamma)\mathbb{E}_{s\sim d_0}[f(s)] \quad (25)\\
&= \gamma\mathbb{E}_{(s,a,s')\sim d_\pi}[w_{\pi/\pi_0}(s)^{-1}w(s)f(s')] - \mathbb{E}_{s\sim d_\pi}[w_{\pi/\pi_0}(s)^{-1}w(s)f(s)] + (1-\gamma)\mathbb{E}_{s\sim d_0}[f(s)] \\
&= \sum_{s'} \delta(g,s')f(s'),
\end{aligned}
$$

where we have defined $g(s) := d_\pi(s)w_{\pi/\pi_0}(s)^{-1}w(s)$. Therefore, $L(w,f) = 0$ for $\forall f$ is equivalent to $\delta(g,s') = 0$ for $\forall s'$, which is in turn equivalent to $g(s) = d_\pi(s)$. Therefore, we have $w(s) = w_{\pi/\pi_0}(s)$ when $0 < \gamma < 1$, and $g(s) \propto d_\pi(s)$, or equivalently, $w(s) \propto w_{\pi/\pi_0}(s)$, when $\gamma = 1$. □

*Proof of Lemma 5.* Note that

$$
\begin{aligned}
\Pi f(s) &= f(s) - \gamma\mathbb{E}_{(s',a)|s\sim d_\pi}[f(s')] \\
&= f(s) - \gamma\mathbb{E}_{(s',a)|s\sim d_{\pi_0}}[\beta_{\pi/\pi_0}(a|s)f(s')].
\end{aligned}
$$

Following the proof of Theorem 4 up to (25), we have

$$
\begin{aligned}
L&(w,f) \\
&= \gamma\mathbb{E}_{(s,a,s')\sim d_{\pi_0}}[(\beta_{\pi/\pi_0}(a|s)w(s) - w(s'))f(s')] + (1-\gamma)\mathbb{E}_{s\sim d_0}[(1-w(s))f(s)] \\
&= \gamma\mathbb{E}_{(s,a,s')\sim d_{\pi_0}}[(\beta_{\pi/\pi_0}(a|s)w(s)f(s')] - \mathbb{E}_{s\sim d_{\pi_0}}[w(s)f(s)] + (1-\gamma)\mathbb{E}_{s\sim d_0}[f(s)] \\
&= -\mathbb{E}_{s\sim d_{\pi_0}}\left[w(s)\left(f(s) - \gamma\mathbb{E}_{(s',a)|s\sim d_{\pi_0}}[\beta_{\pi/\pi_0}(a|s)f(s')]\right)\right] + (1-\gamma)\mathbb{E}_{s\sim d_0}[f(s)] \\
&= -\mathbb{E}_{s\sim d_{\pi_0}}[w(s)\Pi f(s)] + (1-\gamma)\mathbb{E}_{s\sim d_0}[f(s)].
\end{aligned}
$$

Since $L(w_{\pi/\pi_0}, f) = 0$, we have

$$
\begin{aligned}
L(w,f) &= L(w,f) - L(w_{\pi/\pi_0}, f) \\
&= \mathbb{E}_{s\sim d_{\pi_0}}[(w_{\pi/\pi_0}(s) - w(s))\Pi f(s)].
\end{aligned}
$$

□

**Lemma 10.** *For any function $g(s)$, define $\bar{g} = \mathbb{E}_{s\sim d_\pi}[g(s)]$ and*

$$
f_g(s) = \begin{cases}
\mathbb{E}_{\boldsymbol{\tau}\sim p_\pi}[\sum_{t=0}^\infty \gamma^t g(s_t) \mid s_0 = s] & \text{when } 0 < \gamma < 1, \\
\lim_{T\to\infty} \mathbb{E}_{\boldsymbol{\tau}\sim p_\pi}[\sum_{t=0}^T g(s_t) - \bar{g} \mid s_0 = s] & \text{when } \gamma = 1,
\end{cases} \quad (26)
$$

*assuming the limits above exist. Then, when $0 < \gamma < 1$, $f = f_g$ is the unique solution of $g = \Pi f$; when $\gamma = 1$ and $\boldsymbol{T}_\pi$ is irreducible, all the solutions of $g - \bar{g} = \Pi f$ satisfies $f = f_g + \text{constant}$.*

*Proof of Lemma 10.* Consider first the discounted case $\gamma \in (0, 1)$, we have

$$
\begin{aligned}
\Pi f_g(s) &= f_g(s) - \gamma \mathbb{E}_{(s',a)|s \sim d_\pi}[f_g(s')] \\
&= \mathbb{E}[\sum_{t=0}^{\infty} \gamma^t g(s_t) \mid s_0 = s] - \gamma \mathbb{E}_{(s',a)|s \sim d_\pi} \left[ \mathbb{E}[\sum_{t=0}^{\infty} \gamma^t g(s_t) \mid s_0 = s] \right] \\
&= \mathbb{E}[\sum_{t=0}^{\infty} \gamma^t g(s_t) \mid s_0 = s] - \mathbb{E}[\sum_{t=0}^{\infty} \gamma^{t+1} g(s_{t+1}) \mid s_0 = s]] \\
&= \mathbb{E}[g(s_0) \mid s_0 = s] \\
&= g(s).
\end{aligned}
$$

For the uniqueness, assume $g = \Pi f_1$ and $g = \Pi f_2$, and $\delta f = f_1 - f_2$, then $\Pi \delta f = 0$, where

$$
\delta f(s) = \gamma \sum_{s'} \boldsymbol{T}_\pi(s'|s) \delta f(s').
$$

If $0 < \gamma < 1$, we have

$$
\|\delta f\|_\infty = \left\| \gamma \sum_{s'} \boldsymbol{T}_\pi(s'|s) \delta f(s') \right\|_\infty \leq \gamma \|\delta f\|_\infty ,
$$

which implies $\|\delta f\|_\infty = 0$.

For the average reward case $\gamma = 1$, we have

$$
\begin{aligned}
\Pi f_g(s) &= f_g(s) - \mathbb{E}_{(s',a)|s \sim d_\pi}[f_g(s')] \\
&= \lim_{T \to \infty} \mathbb{E}[\sum_{t=0}^{T}(g(s_t) - \bar{g}) \mid s_0 = s] - \mathbb{E}_{(s',a)|s \sim d_\pi}[\mathbb{E}[\sum_{t=0}^{T}(g(s_t) - \bar{g}) \mid s_0 = s]] \\
&= \lim_{T \to \infty} \mathbb{E}[\sum_{t=0}^{T}(g(s_t) - \bar{g}) \mid s_0 = s] - \mathbb{E}[\sum_{t=0}^{T}(g(s_{t+1}) - \bar{g}) \mid s_0 = s] \\
&= \mathbb{E}[g(s_0) - \bar{g} \mid s_0 = s] \\
&= g(s) - \bar{g}.
\end{aligned}
$$

For the uniqueness, assume $g = \Pi f_1$ and $g = \Pi f_2$, and $\delta f = f_1 - f_2$, then $\delta f = \sum_{s'} \boldsymbol{T}_\pi(s'|s) \delta f(s')$, which implies $\delta f = \sum_{s'} \boldsymbol{T}_\pi^n(s'|s) \delta f(s')$, where $\boldsymbol{T}_\pi^n$ is the $n$-step transition probability function. If $\delta f$ is not a constant, there must exists a state $\tilde{s}$ such that $\delta f(\tilde{s}) < \|\delta f\|_\infty$. Since $\boldsymbol{T}_\pi$ is irreducible, there exists a $n > 0$ such that $\boldsymbol{T}_\pi^n(\tilde{s}|s) > 0$. Therefore,

$$
\|\delta f\|_\infty = \left\| \boldsymbol{T}_\pi^n(\tilde{s}|s) \delta f(\tilde{s}) + \sum_{s' \neq \tilde{s}} \boldsymbol{T}_\pi^n(s'|s) \delta f(s') \right\|_\infty < \|\delta f\|_\infty ,
$$

which is contradictory. Therefore, $\delta f$ must be a constant. In fact, functions that satisfies $\delta f = \sum_{s'} \boldsymbol{T}_\pi(s'|s) \delta f(s')$ is called harmonic [17, Lemma 1.16]. $\qquad \square$

*Proof of Theorem 6.* By taking $f_g$ such that $g(s) = \mathbf{1}(s = \tilde{s})$, we have

$$
L(w, f_g) = \mathbb{E}_{s \sim d_{\pi_0}}[(w_{\pi/\pi_0}(s) - w(s))g(s)] = d_\pi(\tilde{s}) - w(\tilde{s}) d_{\pi_0}(\tilde{s}).
$$

We just need to calculate $f_g$, following Lemma 10.

Note that $\boldsymbol{T}_\pi^t(\tilde{s} \mid s) = \mathbb{E}_{\tau \sim p_\pi}[\mathbf{1}(s_t = \tilde{s}) \mid s_0 = s)]$. When $0 < \gamma < 1$, we have

$$
\begin{aligned}
f_g(s) &= \mathbb{E}_{\tau \sim p_\pi}\left[ \sum_{t=0}^{\infty} \gamma^t \mathbf{1}(s_t = \tilde{s}) | s_0 = s \right] \\
&= \sum_{t=0}^{\infty} \gamma^t \boldsymbol{T}_\pi^t(\tilde{s}|s).
\end{aligned}
$$

**Algorithm 1** Main Algorithm (Average Reward Case)

---

**Input**: Transition data $\mathcal{D} = \{s_t, a_t, s'_t, r_t\}_t$ from simulator from the behavior policy $\pi_0$; a target policy $\pi$ for which we want to estimate the expected reward. Denote by $\beta_{\pi/\pi_0}(a|s) = \pi(a|s)/\pi_0(a|s)$.

**Initial** the density ratio $w(s) = w_\theta(s)$ to be a neural network parameterized by $\theta$.

**for** iteration = 1, 2, ... **do**

　　Randomly choose a batch $\mathcal{M}$ of size $m$ from the transition data $\mathcal{D}$, $\mathcal{M} \subset \{1, \dots, n\}$.

　　**Update** the parameter $\theta$ by $\theta \leftarrow \theta - \epsilon \nabla_\theta \hat{D}(w_\theta/z_{w_\theta})$, where

$$\hat{D}(w) = \frac{1}{|\mathcal{M}|} \sum_{i,j \in \mathcal{M}} \Delta(w, s_i, a_i, s'_i) \Delta(w, s_j, a_j, s'_j) k(s'_i, s'_j),$$

　　and $z_{w_\theta}$ is a normalization constant $z_{w_\theta} = \frac{1}{|\mathcal{M}|} \sum_{i \in \mathcal{M}} w_\theta(s_i)$.

**end for**

**Output**: Estimate the expected reward of $\pi$ by $\hat{R}_\pi = \sum_{i=1}^n v_i r_i / \sum_{i=1}^n v_i$, where $v_i = w_\theta(s_i)\beta_{\pi/\pi_0}(a_i, s_i)$.

---

For the average reward case, note that $\bar{g} = \mathbb{E}_{s \sim d_\pi}[\mathbf{1}(s = \tilde{s})] = d_\pi(\tilde{s})$, so

$$f_g(s) = \mathbb{E}_{\boldsymbol{\tau} \sim p_\pi}\left[\sum_{t=0}^\infty \mathbf{1}(s_t = \tilde{s}) - d_\pi(\tilde{s}) | s_0 = s\right]$$

$$= \sum_{t=0}^\infty (\boldsymbol{T}_\pi^t(\tilde{s}|s) - d_\pi(\tilde{s})).$$

Similarly, we take $g(s) = \mathbf{1}(s = \tilde{s})/d_{\pi_0}(\tilde{s})$, and obtain bounds for $\left\|w_{\pi/\pi_0} - w\right\|_\infty$. $\qquad \square$

*Proof of Theorem 7.* Define $r_\pi(s) = \mathbb{E}_{a|s \sim \pi}[r(s, a)] = E_{a|s \sim \pi_0}[\beta_{\pi/\pi_0}(a|s)r(s, a)]$, then

$$R_\pi[w] = \mathbb{E}_{s \sim d_{\pi_0}}[w(s)\beta_{\pi/\pi_0}(a|s)r(s, a)] = \mathbb{E}_{s \sim d_{\pi_0}}[w(s)r_\pi(s)].$$

We consider the average reward case first. Following the definition of the operator $\Pi$ in (17) and the average reward Bellman equation, we have

$$\Pi V^\pi(s) = r_\pi(s) - R_\pi.$$

Following Lemma 10, we have

$$L(w, f) = \mathbb{E}_{s \sim d_{\pi_0}}[(w(s) - w_{\pi/\pi_0}(s))(r_\pi(s) - R_\pi(s))] = R_\pi[w_{\pi/\pi_0}] - R[w] = R_\pi - R_\pi[w].$$

For the discounted case, following the definition of $\Pi$ and the discounted Bellman equation (2), we have $\Pi V_\pi(s) = r_\pi$, which gives

$$L(w, f) = \mathbb{E}_{s \sim \pi_0}[(w_{\pi/\pi_0}(s) - w(s))r_\pi(s)] = R_\pi[w_{\pi/\pi_0}] - R[w] = R_\pi - R_\pi[w].$$

$\qquad \square$

# D  Algorithm Details

Algorithm 1 summarizes our main algorithm for the average reward case, where we approximate the mini-max loss function in (12) using empirical averaging of observed data.

The algorithm for the discounted case follows the same idea, but requires some modification due to the additional term in (15). To handle it in a notionally convenient way, we find it is useful to introduce a dummy transition pair $\{s_{-1}, a_{-1}, s'_{-1}, r_{-1}\}$ at time $t = -1$, for which we define $s'_{-1} = s_0$, $r_{-1} = 0$ and $\Delta(w; \ s_{-1}, a_{-1}, s'_{-1}) := 1 - w(s_0)f(s_0)$. Related, we define an augmented discounted visitation distribution via

$$\tilde{d}_\pi(s) = \gamma d_{\pi,t}(s) + (1 - \gamma)d_{\pi,-1}(s) = (1 - \gamma) \sum_{t=-1}^\infty \gamma^{t+1} d_{\pi,t}(s). \qquad (27)$$

**Algorithm 2** Main Algorithm (Discounted Reward Case)

---

**Input**: Transition data $\mathcal{D} = \{s_t, a_t, s'_t, r_t\}_t$ from the behavior policy $\pi_0$; a target policy $\pi$ for which we want to estimate the expected reward. Denote by $\beta_{\pi/\pi_0}(a|s) = \pi(a|s)/\pi_0(a|s)$. Discount factor $\gamma \in (0, 1]$.

**Augment** the data with dummy data $\{s_{-1}, a_{-1}, s'_{-1}, r_{-1}\}$ for which $r_{-1} = 0$, $s'_{-1} = s_0$ and $\Delta(w; s_{-1}, a_{-1}, s'_{-1}) := 1 - w(s_0)f(s_0)$. Add them to $\mathcal{D}$ to form an augment dataset $\tilde{\mathcal{D}}$.

**Initial** the density ratio $w(s) = w_\theta(s)$ to be a neural network parameterized by $\theta$.

**for** iteration = 1, 2, ... **do**

  Randomly choose a batch $\mathcal{M} \subseteq \{1, \dots, n\}$ from the augmented transition data $\tilde{\mathcal{D}}$, by selecting time $t$ with probability proportional to $\gamma^{t+1}$.

  **Update** the parameter $\theta$ by $\theta \leftarrow \theta - \epsilon\nabla_\theta\hat{D}(w_\theta/z_{w_\theta})$, where

$$\hat{D}(w) = \frac{1}{|\mathcal{M}|} \sum_{i,j\in\mathcal{M}} \Delta(w, s_i, a_i, s'_i)\Delta(w, s_j, a_j, s'_j)k(s'_i, s'_j),$$

  and $z_{w_\theta}$ is a normalization constant $z_{w_\theta} = \frac{1}{|\mathcal{M}|}\sum_{i\in\mathcal{M}} w_\theta(s_i)$.

**end for**

**Output**: Estimate the expected reward of $\pi$ by $\hat{R}_\pi = \sum_{i=1}^n v_i r_i / \sum_{i=1}^n v_i$, where $v_i = w_\theta(s_i)\beta_{\pi/\pi_0}(a_i, s_i)$.

---

Under this notation, the loss (15) of discounted case is rewritten into a form identical to the average reward case:

$$L(w, f) = \gamma\mathbb{E}_{(s,a,s')\sim d_{\pi_0}}[\Delta(w; s, a, s')f(s')] + (1-\gamma)\mathbb{E}_{s\sim d_0}[(1 - w(s))f(s)]$$
$$= \mathbb{E}_{(s,a,s')\sim\tilde{d}_{\pi_0}}[\Delta(w; s, a, s')f(s')].$$

Therefore, following Theorem 2, we have

$$\max_{f\in\mathcal{F}} L(w, f)^2 = \mathbb{E}_{\tilde{d}_{\pi_0}}\left[\Delta(w; s, a, s')\Delta(w; \bar{s}, \bar{a}, \bar{s}')k(s', \bar{s}')\right], \tag{28}$$

when $\mathcal{F}$ is the ball of RKHS with kernel $k(s', \bar{s}')$.

We can further approximate the expectation $\mathbb{E}_{\tilde{d}_{\pi_0}}[\cdot]$ given a set of augmented trajectories $\tilde{\mathcal{D}} = \{s_t, a_t, s'_t, r_t\}_{t=-1}^T$. Following (27), this can be done by randomly drawing (with replacement) data at time $t$ with probability proportional to $\gamma^t$. Let $\{s_t, a_t, s'_t, r_t\}_{t\in\mathcal{M}}$ be a subset of $\tilde{\mathcal{D}}$ generated this way, and the mini-max loss in (28) can be approximated by

$$\max_{f\in\mathcal{F}} L(w, f)^2 \approx \frac{1}{|\mathcal{M}|}\sum_{i,j\in\mathcal{M}} \Delta(w, s_i, a_i, s'_i)\Delta(w, s_j, a_j, s'_j)k(s'_i, s'_j).$$

This equation is identical to the one in Algorithm 1 for the average case, but differs in the way the minibatch $\mathcal{M}$ is generated: it includes the dummy transition at time $t = -1$ with probability $(1 - \gamma)$ and select time $t$ with discounted probability $\gamma^{t+1}$. See Algorithm 2 for the summary of the procedure.

## E   Information on SUMO Traffic Simulator

We provide details of the SUMO traffic simulator and how we formulate it as a standard reinforcement learning problem.

**States for SUMO**   A states of a traffic should provide us with enough information to control the traffic light. A complex way is an image-like representation of the traffic vehicle around the traffic light intersection [41]. Here, to simplify the problem, we add lane detectors around traffic light intersections, and count the total number of vehicles on each lane as states $s_t$. This should give us enough, though not perfect, information to guide the traffic light agent to choose its action.

**Actions**  For a standard crossing intersection, its traffic light will have a program for 8 phases: "Straight signal for North-South", "Turn-left signal for North-south", "Straight signal for East-West", "Turn-left signal for East-west" and their corresponding "yellow light" slow down signals. Here, we simplify these 4 phases into actions $a_t$ for each traffic light, where we let one big time step $t$ in reinforcement learning setting to be 6 real time steps in SUMO simulator. Within each big time step $t$, we add a transition of 3 real time steps "yellow light" phase as a buffer to prevent vehicles for "emergency stop" if our agent decides to change light status ($a_t \neq a_T$).

**Rewards**  Our goal is to minimize the total travelling time for all vehicles. Thus, we could set the negative of current aggregate total number of vehicles during the one big time step as reward $r_t$. To simplify, we can just consider 6 times the current total number of vehicle as a approximation of $r_t$ to make our system simpler.

**Policy**  We use linear policy with the final softmax layer as probability for each action. We train a policy $\pi_*$ using Cross entropy(CE) method for 10 iterations and set it to be the target policy. And we set the policies at the training iteration 6, 7, 8, 9 as behavior policies, which correspond to x-ticks 1-4 in Figure 4(c).

**Other details**  To simulate on our given network, we also need to design route documents for a vehicle to follow. Each route is a set of roads that connect any two exit nodes from the map. To make simple but reasonable routes for the vehicle, we constrain our routes with at most one turn in the network to avoid detours. We control each route with a fixed probability (different from each route) every time step to generate a vehicle, to guarantee a randomized environment.

## Footnotes

[1] This is different from the average reward case, in which $d_\pi(s)$ is the stationary distribution of $\boldsymbol{T}_\pi$.