[Reviews · NeurIPS 2018]

Reviewer 1



This paper introduces off-policy techniques for estimating average rewards. The technique relies on the fact that average reward has a reformulation based on stationary state distribution. Therefore, instead of computing the importance sampling ratio of the trajectory, which has exponentially exploding variance with the horizon, it suffices to compute the importance sampling ratio of the stationary state distribution. Theoretical and empirical results support the technique. I find the work quite interesting. The paper is well written. The work is original with a significant contribution, that is, avoiding multiplicative ratios in one case of multi-step off-policy learning. There are some limitations which might be corrected without drastic changes. On the other hand, the scope of the work is more limited than it appeared initially. Authors' response to these concerns will be highly appreciated and useful. The scope of the technique proposed seems to be limited only to the case of average reward. Much of the work in off-policy learning revolves around what the authors referred to as value-function learning. If this limitation is not clarified up front, the work may leave an impression that it offers to break the curse of the horizon for value-function learning. However, it does not. How this technique applies to value-function learning is not clear. On the other hand, the reformulation of average reward based on stationary state distribution is a known result. In that sense, the overall estimator based on density ratio is quite obvious. Hence, more of the contribution goes here in the particular way the density ratio is estimated here. However, for that, an analysis of computational complexity and providing a pseudo code of the overall technique would have been useful. Can the authors offer some insight into how value-function learning can benefit from this technique? Alternatively, can you shed some light on the possibility of replacing off-policy value-function learning with off-policy average-reward learning altogether? *** After author rebuttal *** Thanks for the clarification that you indeed noted that the paper is about the average reward case. I am not sure the title, which is more general than the contribution, is doing justice to the work. On the other hand, if the extension to the value function learning case is that straightforward, it would have made more sense to start with that first. If the extension is not really that straightforward, it is okay to mention that, which would give more credence to why it was not tried first. For computational complexity, it would also be useful to know what would be the cost for getting w(s) through (11) in the first place. Thanks for the rebuttal and well done!

Reviewer 2



This submission proposes a method for off-policy estimation. The key idea behind the method is to use an importance sampling scheme that includes ratio of state densities rather than the ratio of the probabilities of the trajectories induced by the target and behavior policies. The theoretical properties of this estimator are then presented along with methods to estimate these ratios both for the discounted and average reward case. Experiments are then conduced on three toy environments. Some environements have a discrete state space, others a continuous one. The proposed approach is shown to outform the baselines in significant experimental settings. Overal, I think this paper is a good submission. - Off-policy estimation is an important problem. - The paper is very well-written. I particularly like the motivating example in the appendix and the organisation of the paper that presents all important points in the body of the paper while providing lots of details and proofs in the appendix. While the paper contains lots of theorems (and I could not check them all), the authors explains why these are interesting and the theory flows quite well. - The experiments decently show the strength of the proposed method: three different problems are considered (both discrete and continuous ones) and several other classes of off-policy methods are considered. - The proposed approach comes with theoretical guarantees. - To the best of my knowledge, this has not been done before and opens interesting research diretions. Here are a few points that could be improved. - The experiments might be non-trivial to reproduce. There are several theorems that provide various methods to estimate the weight of the proposed approach. Making available the code of the experiments would greatly help. So would providing some guidelines to choose between the kernel based approach and the neural network one. What motivated this choice in the experiments? - In Figure 4c, how are the different policies selected? What happens for policies closer to the target than policy 4? The current plot suggests that the proposed method might have worse performance than some baselines. It would be interesting to know whether the proposed method might not be the best choice when the two policies are too similar. - only toy problems - It could be useful to include comparison to more state of the art IS methods. ---------------- Thank your for answering my questions.

Reviewer 3



This paper addresses one major limitation of Importance Sampling (IS) estimators for RL tasks, that is its exponentially growing worst case variance with the trajectory length. They name this limitation as "curse of horizon". By considering an alternative definition for IS estimators which is based on the state visitation distribution, the remove the dependence of the estimator in trajectory length. With the new definition, the only difficulty that remains is to estimate the state visitation ratio between evaluation policy and behavior policy. For this, they propose a minimax loss function which simplifies to a closed-form solution under some assumptions. The numerical experiments illustrate the superior performance of the estimator with respect to IS estimators. The paper is well-written. I believe that the contribution of the work is original and significant to the field. In my point of view, the only missing point that it is not well-discussed is how they can replace IS estimators. Following, I have several suggestions/questions that might improve the presentation of the work. 1) I think that several steps are missing including an implementation guide between the theoretical section and experiments. For example: 1-1) How hard is its implementation w.r.t. the IS estimators? It seems that the only extra step is a supervised learning before the RL training, right? 1-2) Once \hat{w} is available, what is the complexity of computing $R(\pi)?$ Better or worse than IS estimators? 1-3) I think that providing an algorithm (maybe in Appendix) which summarizes the steps would help the reader. 2) I would suggest comparing the numerical experiments with doubly-robust estimators (Jiang et al. 2015) as the state-of-the-art off-policy evaluation estimators. The current baselines are sufficient for showing the capability of your proposed method, but one cannot confirm whether they are SOTA or not. 3) How restrictive is it to have $f$ in a unit ball of RKHS (Thm2)? Please elaborate. Minor: In (7), $m$ is not defined beforehand -- it is introduced in Appendix A for the first time. =========== After reading the rebuttal, I found clear explanation to all my questions/comments. Thanks.